# Transfer Learning: Video Prediction and Spatiotemporal Urban Traffic Forecasting †

**Dmitry Pavlyuk**

Transport and Telecommunication Institute, LV-1019, Riga, Latvia; Dmitry.Pavlyuk@tsi.lv

† This paper is an extended version of our paper published in the Proceedings of the 6th International Conference on Models and Technologies for Intelligent Transportation Systems (MT-ITS) (Cracow, Poland, 5–7 June 2019).

**Abstract:** Transfer learning is a modern concept that focuses on the application of ideas, models, and algorithms, developed in one applied area, for solving a similar problem in another area. In this paper, we identify links between methodologies in two fields: video prediction and spatiotemporal traffic forecasting. The similarities of the video stream and citywide traffic data structures are discovered and analogues between historical development and modern states of the methodologies are presented and discussed. The idea of transferring video prediction models to the urban traffic forecasting domain is validated using a large real-world traffic data set. The list of transferred techniques includes spatial filtering by predefined kernels in combination with time series models and spectral graph convolutional artificial neural networks. The obtained models' forecasting performance is compared to the baseline traffic forecasting models: non-spatial time series models and spatially regularized vector autoregression models. We conclude that the application of video prediction models and algorithms for urban traffic forecasting is effective both in terms of observed forecasting accuracy and development, and training efforts. Finally, we discuss problems and obstacles of transferring methodologies and present potential directions for further research.

**Keywords:** urban traffic flows; spatiotemporal models; data-driven; graph convolutional neural networks; spatial filtering; network-wide forecasts

---

## 1. Introduction

Transfer learning (or domain adaptation) is a modern concept defined as application of principles and models, learned in one setting, for improving solutions in another setting — for example, application of models that developed for video prediction for forecasting of urban traffic flows [1,2]. In a machine learning context, transferring is usually implemented via adaptation of a trained model by transforming the input feature space to match domain inputs or by replacing the final layer of a model to produce domain-specific outputs. For example, a link between popular problems of text and image classification can be created by developing translator functions for converting text- and image-specific features into a common feature set [3]. Another example of transfer learning in image processing is a model (e.g., convolutional neural network) that is trained for the classification of animals and further fine-tuned for the classification of other types of objects [4]. In a more general context, transfer leaning is not limited by the application of pre-trained models in another area, but also extends to the application of general model architectures, widely used in one domain, for solving problems in another domain. The practical advantage of transfer learning is widely acknowledged: application of pre-trained models or carefully tested model architectures allows saving resources for model training. Resources for model training include both computational power and training data set collection (which can be limited in some not-data-rich domains). At the same time, transfer learning may sufficiently improve the scientific process of developing and testing

new methodologies; the merging of models and principles from different domains leads to a faster and more focused research process.

One of the key issues for successful transfer learning is the similarity of the output results, input space, and background principles between domains. Although there are several statistical approaches to matching domain-specific features and problems [5], none of them guard against negative transfer effects that appear when information from another domain degrades the performance of a learner from another [1]. Additionally, manual matching of domain-specific features and outputs keep the feature set interpretable, which is highly important for decision makers.

In this paper, we provide successful evidence for merging methodologies of two emerging areas: video prediction and spatiotemporal urban traffic forecasting. Video prediction [6] is a popular problem of computer vision, which is devoted to the generation of future video frames on the base of the previous video stream. Recently, the problem has become extremely popular due to the spreading of autonomous robots and self-driving cars, growing computational power, and methodological advances. Spatiotemporal urban traffic forecasting [7] is another emerging domain, which is focused on the prediction of citywide traffic flows and road network states on the base of historical traffic information on linked roads (spatial dimensions), observed for a specified time period (temporal dimension). The problem of spatiotemporal structure identification and traffic forecasting plays a rising role in transport engineering [8]. We state that, despite different domains, methodologies developed for video prediction and spatiotemporal traffic forecasting and related data structures have a high level of similarity, which allows the transfer of them between domains. Further, we provide a brief overview of the historical development of video prediction and spatiotemporal traffic forecasting methodologies and summarize the links between them. In the experimental part of our study, we consider the transferring of video prediction principles and models to the domain of spatiotemporal traffic forecasting (asymmetric transferring). Several methods of video processing are selected and their performance for urban traffic flow forecasting is demonstrated for a large real-world data set.

## 2. Literature Review

### 2.1. Video Prediction Methodology

Video prediction methodology has roots in digital signal and image processing. The key concept of signal processing is filtering, that implements a convolution in temporal or spatial domains with specifically designed kernels [9]. An image (or one frame of a video stream) is usually coded as a bi-dimensional signal, and a wide range of kernels is developed for its spatial convolution. The list of popular kernels includes the Gaussian kernel for smoothing and restoration, first- and second-order kernels for edge and feature detection, and local adaptive kernels for denoising. In addition, several studies were devoted to the development of data-driven kernels, trained on the basis of temporal information [10,11]. These kernels are widely applied in the practice of image processing and have the perfectly developed theoretical background.

Although convolution with spatial kernels demonstrates good results for image processing, their application to video streams is limited due to the omitting of temporal information—each video frame is considered as an independent image and its segments are predicted using spatial information only. Thus, historically, the problem was mainly formulated not as a pure video prediction, but as image inpainting or as video segment restoration. The three-dimensional (3D) video stream (one temporal and two spatial dimensions) has a rich data structure that can be used for the detection of spatiotemporal features like a motion. At the same time, the processing and prediction of multivariate signals require significant computational power and advanced methodologies, hence, the video prediction problem was not closely addressed until the early 2000s. The development of artificial neural networks (ANN) and hardware improvements contributed to advances of video prediction—Sutskever et al. [12] applied a temporal restricted Boltzmann machine for video denoising, and Verma utilized a feed-forward ANN architecture [13] for pixel-wise video

prediction. The input space for every pixel in the latter model included values of immediate vertical and horizontal neighbors for one or two temporal lags, so technically the model was based on data-driven spatiotemporal convolution kernels. Obviously, the simple architecture of a feed-forward ANN did not allow learning long-term temporal dependencies and complex spatial patterns.

The motion of objects is a practically important spatiotemporal pattern that needs to be recognized for video prediction. Global spatial kernels, applied to all pixels of a video frame, will not be able to learn object-specific movements, hence, several enhancements were introduced to capture the motion. Optical flow [14] is an approach to motion estimation that attempts to calculate a position change for every pixel between two consecutive video frames using differential methods. Due to the aperture problem, the solution of optical flow equations is not unique and usually requires additional conditions (e.g., phase correlation). Despite quite strict preconditions (constant brightness of pixels and smooth transitions), the optical flow method demonstrated good performance for the next-frame video prediction [15,16]. Later, the approach was enhanced by the incorporation of a physical model of the observed process. In many practical applications, the video stream represents a natural process that follows physical laws—for example, satellite imagery represents meteorological processes like cyclone movements and can be used for weather forecasting. The background process can be described by a physical model of fluids, based on advection and Navier-Stokes equations, which help to discover spatiotemporal patterns of the video stream's points and regions. Although the assumption on the presence of a physical theory behind the video stream lacks generalization capabilities due to significant specifics of the domain, this approach is widely used as a baseline for video prediction in other domains.

The recent advances of deep learning models allowed for the reduction of assumptions on background processes for video prediction and learning spatiotemporal patterns directly from data. Mainly, image processing is associated with convolutional ANN (CNN), which simultaneously train multiple spatial kernels for feature extraction. Several consecutive layers of trained spatial kernels are supplemented by pooling layers for dimension reduction, rectifier layers for capturing non-linear dependencies, and a fully connected layers for output production. For video processing, CNN architectures were extended for handling temporal dimensions with recurrent ANN architectures. The long short-term memory (LSTM) network is one of recurrent architectures, which, in conjunction with CNN, is widely used for video prediction. The list of developed deep learning models for video prediction includes PredNet [17], MCNet [18], generative adversarial net-based (GAN) [19], dual motion GAN [20], PredRNN [21], among a dozen other architectures. The key problem for ANN architecture development remains the same—the identification of objects and the spatiotemporal patterns of their motion, and the separate prediction of a video background and moving objects.

Classical spatial kernels and CNN architecture exploit the assumption of a fixed number of spatial neighbors for every pixel (usually defined by the grid structure). This assumption is too restrictive for spatiotemporal patterns in video streams—if a video frame contains several moving objects, then pixels that belong to each object are naturally more related to pixels of this object in previous frames than to pixels of other objects (even located within the spatial neighborhood). Thus, the structure of spatiotemporal relationships is better modeled by a graph than by a grid. The spatiotemporal graph contains vertices (individual pixels or regions on sequential video frames) and edges (relationships between vertices). The extension of CNN architecture that implements a convolution on a graph structure is named graph-based convolutional ANN (GCNN) and began emerging in 2019 [22]. Several GCNN architectures were proposed for video prediction: Bhattacharjee and Das [23] suggested an architecture with spatiotemporal graph convolution and the direction attention mechanism; Li et al. [24] developed a GCNN architecture with separate temporal and spatial routing; Shi et al. [25] proposed a two-stream GCNN architecture for skeleton-based motion on video.

Summarizing the historical outline of video prediction methodology, we concluded that the development of algorithms began from predefined spatial convolutions, utilized information from physical background processes, and have come to data-driven deep learning of complex spatial patterns in the form of spatiotemporal graphs.

## 2.2. Spatiotemporal Urban Traffic Forecasting Methodology

The key distinguishing feature of the spatiotemporal approach to urban traffic forecasting is the simultaneous utilization of information on spatial relationships between traffic flow at distant road segments that appear with temporal delays. Special cases of spatiotemporal models include traffic flow over a road network or temporal growing of congestion (usually in the opposite direction to traffic flow). Initially, the appearance of spatiotemporal relationships was explained by the natural features of traffic flow—vehicles observed at an upstream road segment, after a specific time period, will be observed at the downstream one. Later, the reasoning of spatiotemporal dependencies became more complex and now includes the behavior of informed drivers, the supplementary and competitive nature of road segments, and other reasons.

Historically, spatiotemporal urban traffic modeling began with dynamic models of traffic, inherited from the physics of fluids and the kinematic wave theory. Lighthill and Whitham [26] set up an analogy between liquid flows and traffic flows and demonstrated its utility for traffic flow forecasting. Later, using the same analogy, other physical mechanisms were tested for traffic modeling, e.g., Navier-Stokes equations [27]. At the same time, the analogy between flows of particles and vehicles is too restrictive to explain complex traffic phenomena.

From another perspective, a data-driven approach to traffic forecasting was developed on the basis of time series analysis [28]. As the time series models (like autoregressive integrated moving average, ARIMA) utilize temporal information only, their classical specification does not allow for identifying spatiotemporal relationships. Thus, significant efforts were made to enhance time series models with spatial information. In 1984, Okutani and Stephanedes [29] included adjacent roads into the Kalman filter for a given road segment to capture spatiotemporal relationships and utilize this information for forecasting. Despite a significant potential utility of spatiotemporal information, the following significant step in this direction was made in late 1990s only, when advanced time series and machine learning models were applied.

Time series models were enhanced with spatial information in two ways: incorporation of spatial covariates into classical univariate model specifications (e.g., ARIMA with explanatory variables, ARIMAX) and multivariate time series model specifications (vector autoregression, VAR). Williams [30] suggested the ARIMAX model with spatial explanatory variables, defined on the basis of road connectivity (upstream segments) and cross-correlation of traffic flows. Later, several other statistical approaches for the identification of spatial explanatory variables were suggested [31,32]. Another direction of methodological advances, the multivariate time series models, allows for the simultaneous modeling of many road segments and the data-driven identification of spatiotemporal relationships. The classical specification of VAR, the most popular multivariate time series model, is rarely applied to large spatial segments, due to its enormous number of parameters and the potential problem of overfitting. Thus, several sparse specifications of VAR were suggested based on road connectivity [33], cross-correlation [34], adaptive LASSO [35], among others. Drawing a parallel with video processing methodologies, the VAR model in spatial settings represents a linear pixel-specific data-driven spatial kernel, while the sparse VAR model specifications are based on a graph of spatiotemporal dependencies.

Similarly to video prediction, multiple ANN specifications were tested for urban traffic forecasting. The feed-forward ANN specifications were successfully applied to small spatial segments [36,37], but did not work well for large road segments with complex spatiotemporal patterns. Thus, advanced specifications like state-space ANN [38] and time delay ANN [39] were suggested. These specifications required the explicit specification of the spatial or spatiotemporal graph of dependencies. Later, deep learning models were utilized to learn spatiotemporal relationships in a data-driven way. Huang et al. [40] proposed the deep belief network for spatiotemporal feature learning; Cao [41] utilized the tensor-based convolution for model temporal and spatial relationships; and Liang et al. [42] applied the popular GAN architecture.

Recently developed graph-based convolutional ANN architecture quickly found its application in spatiotemporal traffic forecasting. Cui et al. [43] proposed GCNN with a higher order spatial graph convolution and tested its scalability for a citywide road network. Yu et al. [44] applied GCNN with

spectral and spatial graph-based convolutions. Zhang et al. [45] extended GCNN by the learning of several kernels for every road segment and weighting them for construction of sparse locally connected networks.

Summarizing the development of spatiotemporal urban traffic forecasting methodologies, we state that the two most popular approaches, multivariate time series and machine learning models, have come to the data-driven learning of spatiotemporal graphs of dependencies, and are used for better forecasting performance and interpretability of solutions.

*2.3. Transferring Methodologies between Video Prediction and Spatiotemporal Urban Traffic Forecasting*

The brief discussion of video prediction and spatiotemporal urban traffic forecasting methodologies, presented above, reveals multiple similarities of data structures, emerging problems, and applied models and algorithms in these areas. We summarized these similarities in Table 1.

**Table 1.** Matching video prediction and urban traffic forecasting methodologies.

| Feature | Video Prediction | Spatiotemporal Urban Traffic Forecasting |
|---|---|---|
| Data structure | | |
| Observation | Pixel | Road segment |
| Spatial setting | Video frame | Citywide road network |
| Temporal setting | Sequence of video frames | Sequence of temporally aggregated traffic states |
| Modeled variable | Multiple channels for every pixel (e.g., Red, Green, Blue) | Multiple traffic flow characteristics (e.g., flow value, speed, occupancy) |
| Problem dimension | Huge spatial and temporal dimensions (e.g., 1920 × 1080 resolution of 24 frames per second) | Huge spatial and temporal dimensions (e.g., thousands of road segments in a medium-sized city with 30-s aggregation) |
| Data availability | Data-rich area | Data-rich area |
| Dependencies | Spatiotemporal graph | Spatiotemporal graph |
| Methodology | | |
| Type | Spatiotemporal | Spatiotemporal |
| Forecasting features | Separate prediction of stable regions (backward scene) and dynamic objects (motion) | Separate prediction of normal and abnormal traffic conditions (congestion) |
| Attention | Dynamic objects (recognition and prediction of motion) | Dynamic "objects" (congestion and prediction of its growth) |
| Physical analogies | Physics of observed process (e.g., optical flow models) | Analogy with physics of fluids (e.g., kinematic macroscopic traffic flow models) |
| Potential grouping | Patches (stable regions of a video frame) | Clusters (road segments with similar traffic flows) or reservoirs |
| Emerging approaches | Graph-based convolutional ANN | Graph-based convolutional ANN; Multivariate time series with a graph-based structure of dependencies |

A high level of data structure similarity between video and citywide traffic flows led to the development and application of similar models and algorithms. All of this creates favorable conditions for transfer learning between these applied areas—from the utilization of model specifications and algorithms, tested in one setting, in another area, to the direct application of pre-trained models in both areas. Transfer learning between video prediction and traffic forecasting has significant practical potential. First, a researcher who is working on spatiotemporal traffic forecasting models gains a higher start and higher learning slope by utilizing developed video forecasting models instead of testing a wide set of potential model specifications. Second, there are special cases where both video prediction and traffic flow forecasting are required—for example, a traffic operations center serves city-level traffic forecasting and monitoring video streams from cameras for preventing road accidents, or a vehicle on-board system forecasts traffic flows for routing and predict

video streams for autonomous driving. In such cases, it is more commercially attractive to operate and support one model for both tasks instead of having independent models.

Despite the significant advantages of transfer learning in terms of computational resources and scientific efforts, the number of studies at the intersection of video prediction and spatiotemporal urban traffic forecasting is extremely small. In 2017, Ma et al. [46] and Yu et al. [47] represented urban traffic data in the form of images and applied deep learning models, developed for image processing (with convolutional layers and LSTM components), for their forecasting. Ma et al. [46] represented traffic in a linear spatial setting (an arterial road) in the form of a two-dimensional spatiotemporal contour diagram; Yu et al. [47] utilized geographical coordinates of a citywide road network and traffic state color-coding for representing traffic at a specified time period as an image. Krishnakumari et al. [48] also represented traffic data as an image using geographical coordinates and color-coding traffic states, and tested the direct application of pre-trained image processing models for traffic state prediction (the models were fine-tuned after replacing the last fully connected layer of CNN). Finally, Pavlyuk [49] tested different approaches for representing traffic data in a video-like form and for the further application of existing ANN architectures for video prediction.

## 3. Methodology

This study is devoted to the validation of the concept of transfer learning from video prediction to spatiotemporal urban traffic forecasting areas. We arbitrarily selected popular methods of image and video processing and tested their performance for traffic flow data against state-of-the-art models.

Let us have $k$ spatial locations and $Y_t$ as a $k \times 1$ vector $(y_{1,t}, y_{2,t}, \ldots, y_{k,t})'$ of traffic values at the spatial location $i = 1, \ldots, k$ during the time period $t$. We assumed that the one-period spatial structure is provided in the form of the weighted directed graph, where spatial locations are coded into vertices and relationships between them into edges. There is a wide set of approaches to the definition of the spatial structure [8]; in this study, we utilized travel time in uncongested traffic conditions as a primary relationship measure. In addition, we utilized cross-correlation definition of the spatial structure for the state-of-the-art regularized VAR model. We limited the complexity of the spatial structure by defining a local neighborhood for every spatial location $i$ as

$$NB(i,r) = \{j : d_{ij} < r \text{ and } i \neq j\}, \tag{1}$$

where $d_{ij}$ is a travel time between spatial locations $i$ and $j$, $r$ is an arbitrary selected radius of the neighborhood. The radius is normally defined as a maximum travel time of a direct flow within the road network (without loops and forward-backward movements). For some model specifications, we also simultaneously considered two neighborhoods of different radii.

There are several approaches for the calculation of spatial weights on the basis of observed distances. In this study, we tested two options, the negative exponential decay function $w_e$ and the inverse decay function $w_{1/d}$:

$$w_e = w_{ij,e}(\sigma, r) = \begin{cases} exp\left(-d_{ij}^2/\sigma^2\right), if \ j \in NB(i,r), \\ 0, otherwise, \end{cases} \tag{2}$$

$$w_{1/d} = w_{ij,1/d}(\sigma, r) = \begin{cases} \dfrac{1}{d_{ij}}, if \ j \in NB(i,r), \\ 0, otherwise, \end{cases} \tag{3}$$

where $\sigma^2$ is a decay parameter of $w_e$. The negative exponential decay function $w_e$ corresponds to Gaussian smoothing, widely used for image processing.

### 3.1. Transferred Models

In this study, we tested two approaches of different levels of complexity that are widely used in video processing:

- Spatial filtering by predefined kernels (SpX-model), both pure and in combination with the time series model (SpX-ARIMAX), and,
- Graph-based convolutional ANN (GCNN).

### 3.1.1. Models Based on Spatial Kernels

Spatial filtering by predefined kernels is widely used in image processing due to its simplicity and clear results. Dozens of kernels are developed for image smoothing, sharpening, edge detection, interpolation, image inpainting, and other popular tasks. We selected a regression-based technique [10] that uses spatial kernel values as regressors for image inpainting as a base for our model. Following Ohashi and Torgo [10], we selected two spatial kernels for our experiments, weighted average and standard deviation:

$$avg(i, r, t) = \sum_{j \in NB(i,r)} w_{ij}(\sigma, r) y_{j,t} \tag{4}$$

$$sd(i, r, t) = \sqrt{\frac{1}{|NB(i,r)|} \sum_{j \in NB(i,r)} \left( y_{j,t} - avg(i, r, t) \right)^2} \tag{5}$$

The spatially weighted average is a popular kernel for smoothing, while the standard deviation is a variant of edge detection kernels. The key parameter of spatial filtering is a radius of convolution. Smaller radius values correspond to local area characteristics, and larger values to the characteristics of wider areas, and their combination allows the representation of the spatial dynamics of the process. Thus, for our model specification, we simultaneously utilized kernel values for neighborhoods (1) for two radii, $r_1$ and $r_2$: $NB(i, r_1)$ and $NB(i, r_2)$. The sample neighborhoods are presented in Figure 1; neighborhood radii are tuned by the cross-validation procedure.

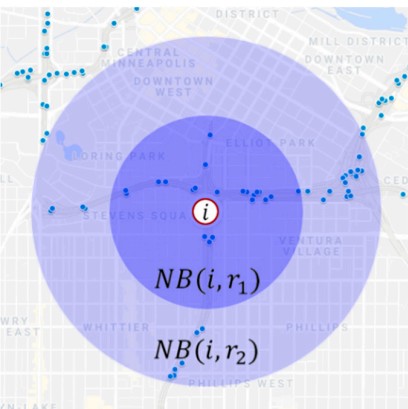

**Figure 1.** Two spatial neighborhoods.

Another potential issue with the definition of the spatial neighborhood is related to the forecasting horizon. Mostly, models are trained for one-step-ahead forecasts and further iteratively applied for longer forecasting horizons. Alternatively, models can be trained for a specified forecasting horizon. For the latter approach to longer forecasting horizons, the spatial neighborhood can be defined on the basis of the forecasting horizon value *h*. Assuming that the acting spatial neighborhood can be wider for longer forecasting horizons, we tested the following specification of expanding spatial neighborhoods:

$$r_h = r + step * h \tag{6}$$

where $step$ is a constant speed of neighborhood expansion ($step = 0$ corresponds to constant spatial neighborhood for all forecasting horizons).

The resulting set of spatial kernels is defined as:

$$SpX_{i,t,h} = \{avg(i, r_{1,h}, t), sd(i, r_{1,h}, t), avg(i, r_{2,h}, t), sd(i, r_{2,h}, t)\} \tag{7}$$

Given the set of regressors $SpX_{i,t}$, we consider the model to predict the traffic flow:

$$y_{i,t+h} = g(\beta, SpX_{i,t,h}) + \varepsilon_{i,t} \tag{8}$$

where $g$ and $\beta$ are the model function and a vector of its parameters, respectively, and $\varepsilon_{i,t}$ is a vector of random terms. Two specifications of $g$ functions are tested: linear and non-linear. The non-linear model specification is implemented using the support vector regression (SVR) [50]. Corresponding specifications are further referred to as SpX-lm (linear) and SpX-SVR (SVR-based model).

Note that SpX-lm and SpX-SVR are based on spatial neighborhoods' states only, and do not include historical information on the spatial location itself. These specifications are fairly restrictive, but allow for transferring models between spatial locations—a model, trained for a spatial location with available historical information, can be applied for the forecasting of traffic flows at another spatial location where the historical information is not available. This approach corresponds to ideas of the cross-region transferring of traffic forecasting models, presented by Lin et al. [51]. In contrast to the mentioned study, our model specification is purely based on traffic data in neighborhoods and does not utilize information on road density, nearby points of interest, and other traffic-specific features.

If historical information on the spatial location is available, then the model specification can be enhanced by combining it with classical time series models. We include spatial regressors $SpX_{i,t}$ into the popular ARIMA model, obtaining the SpX-ARIMAX model specification:

$$y'_{i,t} = \sum_{h=1}^{p} \alpha_h y'_{i,t-h} + \sum_{h=0}^{q} \gamma_h \varepsilon_{k,i,t-h} + \sum_{X \in SpX_{i,t-1,1}} \beta_X X \tag{9}$$

where $y'_{i,t}$ is a stationarized time series $y_{i,t}$, $p, q$ are model orders, and $\alpha_h, \gamma_h, \beta_x$ are model parameters.

### 3.1.2. Models Based on Graph Convolution

While *SpX*-model specification corresponds to well-established and widely used spatial filtering techniques for video processing, the following specification is based on one of the most recent techniques of video prediction—the graph convolutional neural networks (GCNN) [22]. One of the key reasons for the success of conventional CNNs is their ability to learn hierarchical patterns and extract high-level features from image and video data. The GCNN architecture extends CNN by introducing the convolution operator for non-Euclidean graph-based spaces. There are two popular approaches to the implementation of graph-based convolutions:

- Spectral graph convolution [52], and,
- Spatial graph convolution [53].

The spectral graph convolution is based on the spectral graph theory and is calculated via the Laplacian matrix of the graph. Conversely, the spatial graph convolution aggregates information from the local spatial neighborhood of every graph vertex (road segment). Thus, the spectral graph convolution deals with the entire graph are more computationally intensive, while the spatial convolution is local and potentially more effective. In this study, we arbitrarily selected the spectral graph convolution approach for testing.

Let *W* be the matrix of weights that defines the graph structure, and *D* be the diagonal matrix introduced as $D = diag(W\mathbf{1}^T)$, where $\mathbf{1}$ is the all-ones vector. Then, the spectral-based graph convolution is defined as [22].

$$g_\theta * Y = \theta \left( I_k + D^{-\frac{1}{2}} W D^{-\frac{1}{2}} \right) Y, \tag{10}$$

where $I_k$ is an identity matrix, and $g_\theta$ is a filter, represented by the trained matrix $\theta$. Given the graph convolution filter, the architecture of GCNN is similar to classical CNN and includes several sequentially connected convolutional and pooling layers and the final fully connected layer. The general architecture of GCNN is presented in Figure 2.

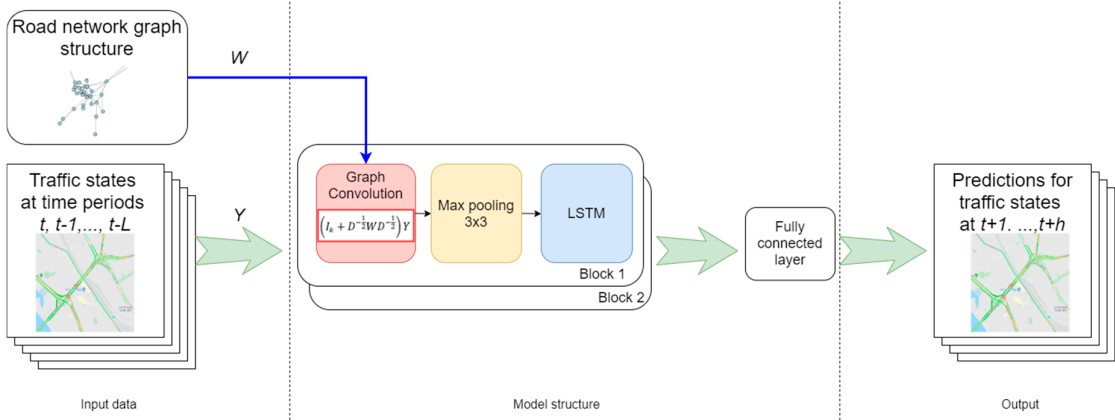

**Figure 2.** General architecture of graph convolutional neural networks (GCNN).

In this study, we utilized the GCNN architecture, developed by Yu et al. [44], as a reference implementation of spectral GCNN and tuned it for our purposes. The resulting architecture includes two sequentially internal blocks of three layers: graph convolution, pooling, and LSTM; other parameters are tuned by cross-validation as described below.

### 3.2. Baseline Models

We utilized several popular traffic forecasting models for the estimation of the comparative performance of the transferred models. The list of baseline models includes the following:

- Naïve forecasts,
- Conventional ARIMA models,
- Conventional VAR models,
- Sparse VAR models in two specifications: with sparsity, controlled using travel time or cross-correlation.

Naïve forecasts and conventional ARIMA models are widely used as univariate (non-spatial) baseline models in traffic forecasting. The conventional VAR model, applied to a vector of traffic values $Y_t$ at spatial locations, is a data-driven approach to learn linear spatial relationships between time series:

$$Y_t = \sum_{h=1}^{p} \Phi_h Y_{t-h} + \varepsilon_t, \tag{11}$$

where $\Phi_h$ is a set of $k \times k$ weight matrices for every lag $h = 1, \ldots, p$.

The conventional VAR specification is extremely flexible and parameter-rich, so for a large dimensionality (citywide traffic forecasting) it suffered from the well-known curse of the dimensionality problem. Sparse VAR specifications, where some coefficients of $\Phi_h$ are forced to be 0, usually demonstrate better forecasting performance [54]. We utilized the sparse specification of VAR, based on local spatial neighborhoods defined above, and introduced it into the model as a set of $S_h$ matrices:

$$s_{ij,h}(r) = \begin{cases} 1, if\ j \in NB(i,r), \\ 0, otherwise, \end{cases} \tag{12}$$

Definition of spatial neighborhoods depends on the distance metric $d_{ij}$. In addition to the travel time-based metric, we utilized the cross-correlation:

$$NB(i,r) = \{j : Corr(y_{i,t}, y_{j,t-h}) > \theta \ and \ i \neq j\}, \tag{13}$$

where $\theta$ is a trained threshold value.

Given the set of regularizing matrices $S_h$, the sparse VAR specification is expressed as

$$Y_t = \sum_{h=1}^{p} (\Phi_h \circ S_h) Y_{t-h} + \varepsilon_t, \tag{14}$$

where $\Phi_h \circ S_h$ is the entry-wise product of the matrices. Further, we refer the sparse VAR specification based on the travel time definition of spatial neighborhoods as SpVAR-tt, and the specification based on cross-correlations, as SpVAR-cc.

## 4. Experimental Results

### 4.1. Data Set

The performance of the discussed models was estimated on the basis of the large real-world data set, obtained from the archive of Minnesota Department of Transportation (MnDoT). The complete data set includes information for 40 weeks (1 January–8 October 2017) from 2676 detectors, deployed in the city center of Minneapolis. Detectors were uniformly distributed over the analyzed roads with a mean distance of 270 m. For calculation purposes, 100 detectors were randomly sampled from the complete data set for further analysis. A map of the case study area with the spatial distribution of the complete data set and the research sample is presented in Figure 3.

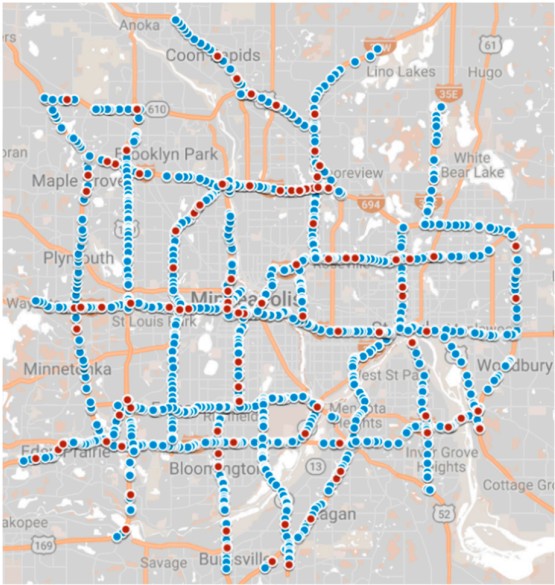

**Figure 3.** Case study area—complete data set (**blue circles**) and research sample (**red circles**).

The original data set includes information on traffic flow volumes, temporarily aggregated by 30-s periods. The executed data preprocessing routines include:

- Lane detectors' values are aggregated by road.
- Values are aggregated in 5-min time intervals, widely used for short-term traffic forecasting.
- Median traffic values are calculated for the first 30 weeks for every 5-min time interval and node, and used as periodical patterns of traffic flows.

- Obtained periodical patterns are subtracted from the flow values for the latest 10 weeks of the data set. As a result, we obtained detrended time series that are used for model training and testing. Thus, the models are focused on the forecasting of deviations from regular traffic conditions (in video prediction, this operation corresponds to the removal of a static background scene).
- Outliers are identified using detector- and time period-specific interquartile ranges. The selected threshold value is selected as 0.01 and is fairly small, so only wrong observations are filtered out, while real traffic values for congested conditions are kept in place. The identified outliers are marked as missed values.
- Linear interpolation is utilized for the imputation of missed values; detectors with more than 4 h of missed values in a row are excluded from the final data set.
- 100 detectors are randomly sampled from the complete data set for computational reasons.

As a result, the preprocessed data set includes a complete 10-week time series (20160 observations) for 100 detectors that represent deviations from regular traffic conditions. The data set is supplemented by a matrix of travel times between detectors, estimated using acting speed limits.

Spatial characteristics of the research sample are presented in Figure 4.

**Figure 4.** Spatial graph settings: (**a**) graph for the 10-min neighborhood; (**b**) percentage of achievable nodes within specified travel time (min); and (**c**) spatial graph Laplacian's heat map, ordered by geographical coordinates.

Figure 4a represents a graph of the spatial neighborhood, constructed for 10-min travel times (unconnected vertices correspond to the entrance points of the research road segment, which are not reachable from other nodes). Figure 4b represents the s-curved distribution of reachable nodes by travel time: percentage of nodes, reachable from a given spatial location within a specified time period under uncongested traffic conditions. The form of the curve closely matches for the complete data set and our research sample, so the characteristics of the sample spatial graph are similar to the complete one. Figure 4c contains a heatmap of the spatial Laplacian matrix, which is used for GCNN convolutions. The graph vertices are ordered by their geographical coordinates for better information on the spatial structure; zero-valued Laplacian rows and columns were excluded from the heatmap for smoother representation of the spatial graph structure.

## 4.2. Hyperparameter Tuning and Forecasting Accuracy

Empirical testing of the research models requires the selection of the performance metric and tuning of the hyperparameters. In this study, we focused on forecasting accuracy as the primary model performance characteristic. The forecasting accuracy is measured using two widely used metrics, mean absolute error (MAE) and root-mean-squared error (RMSE):

$$MAE_t = \frac{1}{k} \sum_{i=1}^{k} |y_{i,t} - \hat{y}_{i,t}|, \tag{15}$$

$$RMSE_t = \sqrt{\frac{1}{k} \sum_{i=1}^{k} (y_{i,t} - \hat{y}_{i,t})^2}, \tag{16}$$

where $\hat{y}_{i,t}$ is a predicted value for a spatial location $i$ and time point $t$. The obtained MAE and RMSE values were aggregated by spatial and temporal dimensions, and their mean values used for model comparisons.

The models' hyperparameters were tuned for one-step-ahead forecasting accuracy, while the model performance was compared for longer forecasting horizons as well. We utilized the rolling window cross-validation technique [55] for the estimation of out-of-sample MAE and RMSE values. The list of tuned hyperparameters includes:

- Radii of spatial neighborhoods $r_1$, $r_2$, and the spatial inflation *step* for SpX-lm, SpX-SVR, and SpX-ARIMAX models;
- Spatial weights' definition and distance decay parameter $\sigma^2$ for SpX and GCNN models;
- ARIMA orders for conventional ARIMA and SpX-ARIMAX models, tuned by Hyndman and Khandakar's algorithm [56];
- Cross-correlation threshold $\theta$ for SpVAR-cc model;
- Order $p$ for VAR, SpVAR-tt, and SpVAR-cc models;
- Size of the rolling window (the look-back interval) $L$ for all models (gradually increased until the models' forecasting performance metrics are stabilized).

The hyperparameters and their tested sets are summarized in Table 2.

**Table 2.** Hyperparameter overview.

| Hyperparameter | Symbol | Tested Values | Used in Models |
| --- | --- | --- | --- |
| Radii of spatial neighborhoods | $r_1$, $r_2$ | [0, 10, 20, 30] | SpX-lm, SpX-SVR, SpX-ARIMAX |
| Spatial inflation | $step$ | [0, 5] | SpX-lm, SpX-SVR, SpX-ARIMAX |
| Spatial weights | $w$ | $w_e, w_{1/d}$ | SpX-lm, SpX-SVR, SpX-ARIMAX, GCNN |
| Distance decay speed for $w_e$ | $\sigma^2$ | [10, 20] | GCNN |
| Cross-correlation threshold | $\theta$ | [0.1, 0.2, 0.3] | SpVAR-cc |
| Order of autoregression and moving average components | $p, q$ | Hyndman and Khandakar's algorithm [56] | ARIMA, SpX-ARIMAX |
| Order of autoregression | $p$ | [1, 3, 6] | VAR, SpVAR-tt, SpVAR-cc |
| Look-back interval | $L$ | [360, 720, 1440] | All models |

The GCNN model was trained for 100 epochs and 3 × 3 windows for spatial and temporal convolutions. In addition, we tested the performance of the GCNN model for detrended and original time series and came to the preference of detrended data, which were also used in all other model specifications.

*4.3. Estimation Results*

The average performance of each model was estimated by the rolling window technique for 10 weeks of data. Given that the first five days were reserved for the look-back interval and the rolling

step was selected as one hour ($l = 12$), we trained every model specification for 1560 data subsets and obtained their short-term forecasts ($h = 3$). Further, the optimal set of hyperparameters for every model specification was selected on the basis of the average one-step-ahead forecasting accuracy.

Obtained forecasting accuracy metrics for the optimal set of hyperparameters are summarized in Table 3 (model specifications with highest forecasting accuracy are marked by bold).

**Table 3.** Forecasting accuracy of the research models.

| Model | Calibrated Hypermeters' Values | MAE by Forecasting Horizon | | | RMSE by Forecasting Horizon | | |
|---|---|---|---|---|---|---|---|
| | | 0–5 min ($h = 1$) | 5–10 min ($h = 2$) | 10–15 min ($h = 3$) | 0–5 min ($h = 1$) | 5–10 min ($h = 2$) | 10–15 min ($h = 3$) |
| | | Transferred models | | | | | |
| SpX-lm | $r_1 = 10$ $r_2 = 30$ $step = 5$ | 11.27 | 11.68 | 11.90 | 16.42 | 17.04 | 17.41 |
| SpX-SVR | $r_1 = 10$ $r_2 = 30$ $step = 5$ | 10.54 | 10.83 | 11.08 | 15.78 | 16.17 | 16.51 |
| SpX-ARIMAX | $r_1 = 10$ $r_2 = 30$ $step = 5$ $w = w_{1/d}$ | **8.85** | 9.56 | 9.91 | 12.54 | 13.75 | 14.33 |
| GCNN | $w = w_{1/d}$ | 9.77 | 10.57 | 11.16 | 18.62 | 23.37 | 25.95 |
| | | Baseline models | | | | | |
| SpVAR-tt | - | 8.92 | 9.42 | 9.84 | 12.58 | 13.40 | 14.09 |
| SpVAR-cc | $\theta = 0.1$ | 8.88 | **9.35** | **9.80** | **12.51** | **13.27** | **14.02** |
| VAR | $p = 6$ | 12.61 | 12.66 | 12.68 | 17.40 | 17.57 | 17.72 |
| ARIMA | detector-specific $p$, $q$ | 9.02 | 9.80 | 10.57 | 12.74 | 14.09 | 15.56 |
| Naïve | - | 16.21 | 16.59 | 16.83 | 25.37 | 25.85 | 26.26 |

Table 3 provides average values of forecasting accuracy metrics, which could be misleading due to the significant heterogeneity of results over spatial and temporal dimensions. Figure 5 represents spatial and temporal kernel densities of MAE values for three model specifications: ARIMA, SpX-ARIMAX, and SpX-SVR (corresponding distributions for SpVAR-tt and SpVAR-cc models were omitted due to their high similarity to the SpX-ARIMAX model).

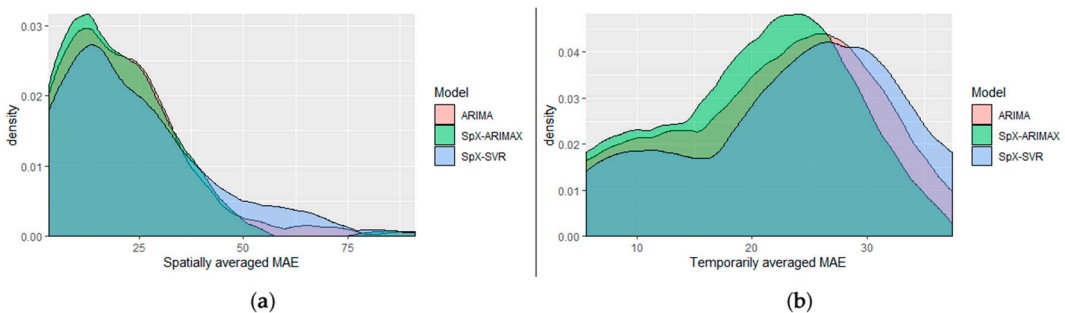

**Figure 5.** Density of mean absolute error (MAE) values: (**a**) spatially aggregated and (**b**) temporarily aggregated.

Spatial distributions were calculated by the averaging of the obtained time-specific MAE values by detectors, while temporal distributions were calculated by averaging the detector-specific MAE values over time of the day.

### 4.4. Reproducibility

To ensure the reproducibility of the obtained experimental results, we publicly provided the source codes for all executed routines at http://bit.ly/Algorithms2020 (R language markdown). The repository includes routines for downloading data from Dr. Kwon's archive of MnDoT traffic data [57], data preprocessing and sampling, and all experiments with model specifications.

## 5. Discussion

Table 3 and Figure 5 demonstrate the comparative forecasting performance of transferred and baseline models. We observed that the overall short-term forecasting performance of the transferred models was effective and comparable with the values of modern spatiotemporal traffic forecasting models. The non-spatial ARIMA model set up a good baseline for one-step-ahead forecasting performance (MAE = 9.02 for $h$ = 1), but its performance was degrading fast for longer forecasting horizons due to its inability to utilize spatial information from neighboring road segments. All spatiotemporal model specifications, except GCNN, demonstrated better stability of performance.

The performances of SpX-lm and SpX-SVR models, based on spatial kernels, were the worst among the analyzed spatial model specifications (MAE = 11.27 for SpX-lm and 10.54 for SpX-SVR), but were still reasonable (e.g., better than the performance of the conventional VAR model specification, MAE = 12.61) Although these performance values were not optimal, the SpX-lm and SpX-SVR models had a significant advantage in terms of transferring—their specifications were based purely on the spatial information and did not include temporal information from the analyzed road segment. This means that the obtained models could potentially be transferred to traffic forecasting at spatial locations, where historical information is not available. The SpX-ARIMAX, which includes both temporal and spatial information, demonstrated one of the best performance values among all models (MAE = 8.85). Note that spatial information was included in the SpX-models in a very simple way—by two predefined spatial kernels—and this definition was good enough to successfully compete against the more detailed definition of the spatial structure, applied in SpVAR-models. Comparing SpX-lm and SpX-SVR model specifications, we observed a clear advantage of the SVR-based model, which indicates non-linear patterns of spatial relationships. These non-linearities were partially utilized by the simultaneous usage of two spatial neighborhoods of different radii, $r_1$ and $r_2$. The radii were tuned by cross-validation and set to 10 and 30 min of travel time for all model specifications. They gradually increased with $step$ = 5 or longer forecasting horizons (e.g., for $h$ = 3 the radii were equal to 20 and 40 min, respectively). Spatial and temporal distributions of MAE values, presented in Figure 5, also indicate the advantages of the SpX-model specifications: comparing the right tails of the distributions (extreme forecasting errors), we observed that the SpX-ARIMAX model provided significantly smaller values. For spatial distribution, it means that the SpX-ARIMAX model was appropriate for all detectors in the research area, while the ARIMA model worked worse for some of them. For temporal distribution, the implication was even more important—the SpX-ARIMAX model gave more stable results for all time periods, including free traffic flow at nights, regular traffic conditions during the day, and congested traffic conditions during peak hours. Overall, we concluded that the approach, based on predefined spatial kernels, widely used for image and video processing, works well for spatiotemporal traffic forecasting.

The direct application of the spectral GCNN model for traffic forecasting was not so successful. The performance of the GCNN model was appropriate for one-step-ahead forecasts (MAE = 9.77) but very unstable in terms of extreme error values (RMSE values were significantly higher, given comparable MAE values). There were several possible explanations for these results. First, the conventional GCNN architecture was used for our experiments, so any traffic-specific features were not taken into consideration (e.g., road capacity, weather conditions, etc.). Second, the training process of GCNN was also specific to video prediction and differed from the other models. For example, the effective look-back interval is limited to several previous frames, which is appropriate for video streams, but could be insufficient for traffic data with longer temporal patterns. Third, our results could be sample-specific (arterial roads), as there is a collection of evidence in the literature that indicates that the GCNN model works well for traffic data in different spatial settings [43–45].

Finally, the reason could alternatively be related to the spectral convolution the graph, while another approach, spatial convolution, could be more appropriate for traffic data. The GCNN methodology itself is relatively new; hence, all these hypotheses require additional research.

Transferring ideas, models, and algorithms between applied areas of video prediction and traffic forecasting opens a wide range of future research directions. Methodological advances that are now implemented in these applied areas in parallel, should be merged and tested for data sets of different natures. In this study, we tested only a limited set of video processing ideas and models, while the modern modeling toolbox is rich and many other exiting models could be adopted. We also considered only one transfer direction, from video prediction to traffic forecasting, while the opposite transfer (application of popular traffic forecasting models to video prediction) is also available. Finally, we presented transferring at two deeper levels, utilizing the same ideas and the same model architectures, while upper-level transfer learning and the application of pretrained models in another application area has a huge research potential and practical utility.

## 6. Conclusions

This study promoted the idea of transfer learning between video prediction and spatiotemporal urban traffic forecasting areas. The transferring of ideas, algorithms, model structures, and pre-trained models is an effective way of methodological enhancement, which allows the saving of computational resources in the practical aspect, and of intellectual and research resources in the scientific aspect. We identified similarities between data structures of video stream and citywide traffic data and discovered close links in historical development and the modern states of video prediction and urban traffic forecasting methodologies. The experimental part of the study was devoted to the application of popular video processing techniques (spatial filtering and convolutional networks) in the traffic forecasting domain. Experiments, executed for the citywide traffic data set, supported our hypothesis on the applicability of video prediction tools for urban traffic data.

**Funding:** The author was financially supported by the specific support objective activity 1.1.1.2. "Post-doctoral Research Aid" (Project id. N. 1.1.1.2/16/I/001) of the Republic of Latvia, funded by the European Regional Development Fund. Dmitry Pavlyuk's research project No. 1.1.1.2/VIAA/1/16/112 "Spatiotemporal urban traffic modelling using big data".

**Acknowledgments:** We thank Taek Kwon for his public archive of MnDoT traffic data [57]. We also thank Yu, Yin, and Zhu for publicly available source codes for the spectral GCNN [44], which we used as a reference implementation.

**Conflicts of Interest:** The authors declare no conflict of interest. The funders had no role in the design of the study; in the collection, analyses, or interpretation of data; in the writing of the manuscript, or in the decision to publish the results.

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
