# Peer review of "Transfer Learning: Video Prediction and Spatiotemporal Urban Traffic Forecasting†"

_algorithms, doi:10.3390/a13020039_

Round 1

Author Response

>> This paper proposed a transfer learning method in video prediction and spatiotemporal traffic forecasting. The idea of transferring video prediction models to the urban traffic forecasting domain is validate using a large real-world traffic data set. The conclusion is that application of video prediction models and algorithms for urban traffic forecasting is effective both in terms of observed forecasting accuracy and development and training effort. The novelty of this work is remarkable, and this paper is well written and organization. It can be accepted with minor revision,

Thank you for your review and valuable comments; our responses are presented below.

>> Comment1: I suggest using r_1, r_2 instead of r1,r2;

Response 1:
r1 and r2 are renamed to r_1, r_2 (subscripts) everywhere as recommended. Additional explanation for these radii is added at lines 276-279; Figure 1 is also updated (line 280).

>> Comment2: Figure 4 (c) is not clear to show you result, please improve it.

Response 2:

Figure 4(c) is slightly updated – zero-valued rows and columns of Laplacian were excluded from the heatmap for better representation of the spatial graph structure. Corresponding notes are added to the paper text (lines 414-417).

>> Comment3: The information in Figure 2 is incomplete, please redraw it.

Response 3:

Figure 2 is redrawn; more detailed information is added.

In addition, English language of the paper text is professionally edited.

Reviewer 2 Report

This paper presents the idea of transfer learning between video prediction and spatiotemporal urban traffic forecasting areas. Transferring of ideas, algorithms, model structures, and pre-trained models is an effective way of methodological enhancement, which allows saving computational resources in the practical aspect and intellectual and research resources in the scientific aspect.

Similar approach to transfer learning was recently applied by Lin et al. [51]. Therefore, can the author explain the main contributions of this paper by comparing it with paper [51]? Can the simulation results can shown comparison between your architecture and Lin et al. [51]?

Lines 238 and 239, r is an arbitrary selected radius of the neighborhood. Lines 263 and 264, Figure 1 shows kernel values for two radiuses – r1 and r2. Can you explain a relation between r, r1, and r2? Figure 1 should show value r to detail.

Figure 4.b represents the s-curved distribution of reachable nodes by travel time. How to calculate % of reachable nodes?  Does it mean longer travel time, the video prediction is better?

In Figure 2, the author utilized the GCNN architecture. Where is this GCNN architecture used in Figure 3? complete data set (blue circles) or research sample (red circles)?

It is easy to understand if the author gives an example which uses the idea of transfer learning between video prediction and spatiotemporal urban traffic forecasting areas.

Author Response

>> This paper presents the idea of transfer learning between video prediction and spatiotemporal urban traffic forecasting areas. Transferring of ideas, algorithms, model structures, and pre-trained models is an effective way of methodological enhancement, which allows saving computational resources in the practical aspect and intellectual and research resources in the scientific aspect.

Thank you for your review and valuable comments; our responses are presented below.

>> Comment1: Similar approach to transfer learning was recently applied by Lin et al. [51]. Therefore, can the author explain the main contributions of this paper by comparing it with paper [51]? Can the simulation results can shown comparison between your architecture and Lin et al. [51]?

Response1:

The similarity between our SpX-SVR model specification and models in Lin et al. [51] relates only to the potential opportunity of forecasting traffic flows at road segments without historical information (and, therefore, of transferring models between road segments). All other aspects of the models are different, namely:

We use only traffic flow information (both as dependent and explanatory variables), while Lin et al. use only traffic-related features (road density, POI) as explanatory variables for traffic flows. Comparison of these approaches seems interesting in terms of traffic forecasting methodologies, but it is not covered in our paper: we focus on transferring methodologies from video prediction domain, where additional features (except the video stream itself) are not natural. We utilize spatial convolution (which corresponds to spatial filters in the image processing domain) of traffic data, while Lin et al. use raw spatial and temporal features. We directly estimate the model for forecasting traffic flows at a target road segment, while Lin et al. forecast traffic flows at nearby segments with available traffic information and construct the target forecast by weighted averaging.

Also, please note that SpX-SVR model specification is only a small part of our research methodology, which also includes more advanced SpX-ARIMAX and GCNN.

Corresponding notes were added to the paper text (lines 302-306) to avoid this confusion of the Reader.

>> Comment2: Lines 238 and 239, r is an arbitrary selected radius of the neighborhood. Lines 263 and 264, Figure 1 shows kernel values for two radiuses – r1 and r2. Can you explain a relation between r, r1, and r2? Figure 1 should show value r to detail.

Response2:

r1 and r2 (now renamed to r_1, r_2, subscripts) are just two specific values of r – so we utilize values of spatial kernels for two neighborhoods,  and . Simultaneous usage of two neighborhoods allows capturing spatial dynamics of traffic flows.

Corresponding notes are added to the paper text (lines 276-279); Figure 1 is also updated in respect to this comment.

>> Comment3: Figure 4.b represents the s-curved distribution of reachable nodes by travel time. How to calculate % of reachable nodes?  Does it mean longer travel time, the video prediction is better?

 Response3:

% of reachable nodes is calculated as an average of the ratio: number of nodes, which can be reached from a given location within a specified travel time (X axis) under uncongested traffic conditions / total number of nodes within the network. This metric represents a structure of the road network (segment size, connectivity, etc.) and is not related to forecasting results. In addition, we used this curve to compare the graph structure of the complete network and our sample (it reveals their similarity).

Corresponding notes are added to the paper text (lines 408-411).

>> Comment4: In Figure 2, the author utilized the GCNN architecture. Where is this GCNN architecture used in Figure 3? complete data set (blue circles) or research sample (red circles)?

Response4:

All model specifications (including GCNN) are tested on the research sample (read circles in Figure 3). We utilize sampling instead of experiments with the complete data set for computational reasons (mentioned in the line 395)

>> Comment5: It is easy to understand if the author gives an example which uses the idea of transfer learning between video prediction and spatiotemporal urban traffic forecasting areas.

Response5:

The following discussion is added to the paper text (lines 210-218):

Transfer learning between video prediction and traffic forecasting has a significant practical potential. First, a researcher who is working on spatiotemporal traffic forecasting models gains a higher start and higher learning slope by utilizing developed video forecasting models instead of testing a wide set of potential model specifications. Second, there are special cases where both video prediction and traffic flow forecasting are required – for example, a traffic operations center serves city-level traffic forecasting and monitoring video streams from cameras for preventing road accidents or a vehicle on-board system forecasts traffic flows for routing and predict video streams for autonomous driving. In such cases this is more commercially attractive to operate and support one model for both tasks instead of having independent models.

Round 2

Reviewer 2 Report

All my comments have been addressed in a corresponding manner. I have no other comments.